# Modeling and Analysis of KSnI_3_ Perovskite Solar Cells Yielding Power Conversion Efficiency of 30.21%

**DOI:** 10.3390/nano15080580

**Published:** 2025-04-11

**Authors:** Bonginkosi Vincent Kheswa, Siyabonga Ntokozo Thandoluhle Majola, Hmoud Al-Dmour, Nolufefe Muriel Ndzane, Lucky Makhathini

**Affiliations:** 1Department of Physics, University of Johannesburg, 55 Beit Street, Doornfontein 2028, South Africa; sntmmajola@gmail.com; 2Department of Physics, Faculty of Science, Mu’tah University, Mu’tah 61710, Jordan; hmoud79@mutah.edu.jo; 3Academic Development Centre, University of Johannesburg, 55 Beit Street, Doornfontein 2028, South Africa; fndzane89@gmail.com; 4Department of Physics and Astronomy, University of the Western Cape, P/B X17, Bellville 7535, South Africa; lmakhathini@uwc.ac.za

**Keywords:** KSnI_3_, rGO, SnO_2_, perovskite, solar

## Abstract

KSnI_3_-based perovskite solar cells have attracted a lot of research interest due their unique electronic, optical, and thermal properties. In this study, we optimized the performance of various lead-free perovskite solar cell structures—specifically, FTO/Al–ZnO/KSnI_3_/rGO/Se, FTO/LiTiO_2_/KSnI_3_/rGO/Se, FTO/ZnO/KSnI_3_/rGO/Se, and FTO/SnO_2_/KSnI_3_/rGO/Se, using the SCAPS-1D simulation tool. The optimization focused on the thicknesses and dopant densities of the rGO, KSnI_3_, Al–ZnO, LiTiO_2_, ZnO, and SnO_2_ layers, the thickness of the FTO electrode, as well as the defect density of KSnI_3_. This yielded PCE values of 27.60%, 24.94%, 27.62%, and 30.21% for the FTO/Al–ZnO/KSnI_3_/rGO/Se, FTO/LiTiO_2_/KSnI_3_/rGO/Se, FTO/ZnO/KSnI_3_/rGO/Se, and FTO/SnO_2_/KSnI_3_/rGO/Se perovskite solar cell configurations, respectively. The FTO/SnO_2_/KSnI_3_/rGO/Se device is 7.43% more efficient than the FTO/SnO_2_/3C-SiC/KSnI_3_/NiO/C device, which is currently the highest performing KSnI_3_-based perovskite solar cell in the literature. Thus, our FTO/SnO_2_/KSnI_3_/rGO/Se perovskite solar cell structure is now, by far, the most efficient PSC design. Its best performance is achieved under ideal conditions of a series resistance of 0.5 Ω cm^2^, a shunt resistance of 10^7^ Ω cm^2^, and a temperature of 371 K.

## 1. Introduction

Perovskite solar cells (PSCs) are an emerging class of photovoltaic solar cells, in which materials known as perovskites are used as an active absorption layer. These materials have a unique crystal structure of the form ABX_3_, where the A-site is occupied by larger cations, which can be organic molecules such as formamidinium or inorganic ions such as Cs [1]. The B-site is typically filled with a metal cation, such as lead or tin, while the X-site is filled with a halide ion such as iodine. This distinctive crystal arrangement gives perovskites their remarkable electronic and optical properties, such as high light absorption, fast charge-carrier mobility, and adjustable band gaps. Thus far, PSCs have shown outstanding improvement in their power conversion efficiency, with some laboratory prototypes even outperforming traditional silicon-based solar cells. Furthermore, their high efficiency presents a unique opportunity for large-scale and cost-effective manufacturing.

There is a wide range of PSCs based on different perovskite materials. Lead-based perovskite solar cells have been the most promising, achieving power conversion efficiencies (PCEs) of up to more than 25% in a laboratory environment [2]. In fact, they are still an active area of research and have been showing improvement in various aspects in numerous computational and experimental studies (see Refs. [3,4,5,6,7,8,9,10,11] and references therein). For instance, the recent study of Ref. [3] investigated a formamidinium lead iodide-based PSC using GPVDM software version 7.88 and obtained an improved PCE of 27.49%, which was also significantly higher than the PCE of the lead-free formamidinium tin iodide PSC that was also explored in the same study. Ref. [4] studied CsPbX_3_-based perovskite solar cells using density functional theory (DFT) and SCAPS-1D (Solar Cell Capacitance Simulator) software version 3.3.11 and found CsPbIBr_2_ to have the best balance between stability and band gap, yielding a maximum PCE of 16.53%. In the same vein, Ref. [5] investigated HTM (hole transport material)-free CsPbI_3_/CsSnI_3_ heterojunction solar cells, using the SCAPS-1D simulation tool, and achieved PCEs of up to 19.99%. Similarly, in the research of Ref. [6], thickness optimization engineering of the electron transport, hole transport, and perovskite layers of MAPbI_3_-based PSCs was performed using the SCAPS-1D simulation package. The study revealed the necessity to increase the layer thickness by 50 to 100% and the improvement of the PCE by 1.5%, achieving a PCE of 22.10%. Furthermore, in the study of Ref. [7], CsPbI_3_, FAPbI_3_, MAPbI_3_, and FAMAPbI_3_ PSCs were analyzed and optimized using SCAPS-1D and results yielded the highest PCE of 26.35% observed on FAMAPbI_3_-based perovskite solar cells. In the same interest, Ref. [8] conducted a numerical study on CsPb._625_Zn._375_IBr_2_-based perovskite solar cells by optimizing the density of charge carriers, the density of defects, and thicknesses of the electron transport layer (ETL), hole transport layer (HTL), and active absorption layer. This yielded a PCE of 21.05% for the optimized structure. Furthermore, Ref. [9] performed a computational study of various perovskite solar cells, offering a detailed investigation of the impact of critical properties, such as band gap, electron affinity, layer thickness, absorption, recombination rate, band alignment, and defects, on the performance, and thus achieved a maximum PCE of 22.05% for the FAPbI_3_-based device. Ref. [10] investigated the effect of incorporating an interfacial layer of BiI_3_ in MAPbI_3_ and MAGeI_3_-based perovskite solar cells and improved the PCE from 19.28 to 20.30% for the MAPbI_3_ PSC.

Even though lead-based PSCs are such a highly promising improvement in the photovoltaic industry, they have tremendous drawbacks. In particular, lead is a very toxic element that poses health risks to the manufacturers and consumers of lead-based perovskite solar cells. Thus, a lot of computational and experimental research is currently devoted to the development and optimization of lead-free perovskite solar cells [3,12,13,14,15,16,17,18,19,20,21]. One such lead-free PSC is the KSnI_3_-based device, which is the subject of interest in the present study. This has recently attracted enormous interest globally, due to the promising structural stability, thermal stability, mechanical stability, optical properties, and electronic properties of KSnI_3_ perovskite material, making it ideal for usage as the active absorption layer in various PSC configurations [22]. In particular, the recent DFT study of Ref. [22] showed KSnI_3_ has (i) tolerance and octahedral factors of 0.803 and 0.531, respectively, proving structural stability; (ii) negative formation energy of −3.6 eV, reflecting thermal stability of the orthorhombic phase; (iii) elastic constants that satisfy conditions for mechanical stability; (iv) low values of reflectivity (0.348 to 0.197) and loss factor (0.065 to 0.005) as well as high values of absorption coefficient (10^6^ to 10^5^ cm^−1^) in the visible light energy region; and (v) electron and hole mobilities of 0.002128 m^2^/Vs and 0.00194 m^2^/Vs, respectively, which are comparable to those of other lead-free perovskites. Thus, KSnI_3_ has potential usage as the active absorption layer in perovskite solar cells using SCAPS-1D, and Ref. [22] obtained the highest PCE of 9.776% on the FTO/TiO_2_/KSnI_3_/Spiro-OMeTAD/W device. In the study of Ref. [23], the impact of organic charge transport layers was explored, and critical parameters such as dopant density, thickness, and defect density were optimized, yielding the highest PCE of 10.83% for an FTO/C_60_/KSnI_3_/PTAA/C structure. In the same vein, the theoretical study of Ref. [24] investigated the impact of metal phthalocyanines charge transport layers on KSnI_3_-based perovskite solar cells and obtained the optimized FTO/F_16_CuPc/KSnI_3_/CuPc/C architecture with a PCE of 11.91%. Another computational study was carried out by Ref. [25] using SCAPS-1D, exploring the effect of charge transport materials on KSnI_3_-based PSCs, and achieved a PCE of 9.28% on the FTO/ZnOS/KSnI_3_/NiO/C configuration. Furthermore, the computational study of Ref. [26] made significant progress on the optimization of KSnI_3_ perovskite solar cells by achieving a PCE of 20.99% for the FTO/ZnO/KSnI_3_/CuI/Au configuration, through the optimization of the hole transport layer, electron transport layer, anode material, and defect and dopant densities. In the same interest, the very recent research of Ref. [27] optimized charge transport layers and incorporated a buffer layer in KSnI_3_ PSCs perovskite solar cells and obtained a maximum PCE of 22.78% for the optimized FTO/SnO_2_/3C–SiC/KSnI_3_/NiO/C structure, which is currently the highest performance ever achieved on KSnI_3_-based PSCs. This PCE value is much lower than the Shockley–Queisser limit of 34%, which is the theoretical PCE limit of solar cells. This limit is mainly due to (i) radiative recombinations whereby electron–hole pairs recombine before contributing to photo-current and (ii) energy dissipation to the crystal lattice whereby electrons that are excited to high energy levels of the conduction band de-excite to the bottom of the conduction band and dissipate the excess energy as heat in the lattice. On the other hand, the recent computational optimization studies of Refs. [28,29,30] recently achieved a PCE of 31% for CsSnI_3_-based PSCs and 27% and 31.62% for CsSnBr_3_-based PSCs when using rGO and WSe_2_ transport layers, respectively, which have never been investigated on KSnI_3_-based PSCs. Thus, there is still hope that the PCE of KSnI_3_-based PSCs can still be significantly improved using carefully chosen optimized HTL and ETL materials, reducing the impacts of the aforementioned energy loss mechanisms.

In this research, we performed computational optimization of FTO/Al–ZnO/KSnI_3_/rGO/Se, FTO/LiTiO_2_/KSnI_3_/rGO/Se, FTO/SnO_2_/KSnI_3_/rGO/Se, and FTO/ZnO/KSnI_3_/rGO/Se, using the state-of-the-art SCAPS-1D simulation tool. Our choice of Al–ZnO, LiTiO_2_, ZnO, SnO_2_, and rGO was driven by their high charge mobilities, strong thermal stabilities except for the case of ZnO, excellent conductivities, and optimal band alignments with KSnI_3_ [29,30,31,32,33,34]. Furthermore, rGO is known for having low trap states in the absorption layer/hole transport layer interface [35]. In particular, we optimized the aforementioned structures by varying the thickness of each layer, the dopant densities of ETLs, KSnI_3_, and rGO layers, and the defect density of KSnI_3_. This interestingly yielded tremendous improvement in the performance of KSnI_3_-based perovskite solar cells. This paper is structured as follows. Section 2 provides the details of the computational methods used in this study. Section 3 presents our results and their comparison with the literature, while Section 4 contains the conclusions.

## 2. Computational Methods

### 2.1. Model Structure and Simulation Tools

The numerical modeling and optimization of perovskite solar cells can be achieved using various simulation tools, such as wxAMPS version 2.0, PC-1D version 5.9, AFORS-HET version 2.5, Silvaco TCAD version 1.24.0.R, OghmaNano version 8.0, and SCAPS-1D version 3.3.11 [36]. In the present study, we focused on the study of four perovskite solar cells using the SCAPS-1D simulation software. In particular, we optimized the performance of FTO/Al-ZnO/KSnI_3_/rGO/Se, FTO/LiTiO_2_/KSnI_3_/rGO/Se, FTO/ZnO/KSnI_3_/rGO/Se, and FTO/SnO_2_/KSnI_3_/rGO/Se PSC devices by tuning their dimensions and electronic properties within the SCAPS-1D numerical tool, which was chosen based on its reliability when compared to experimental data [37] and user-friendliness. The SCAPS-1D software package was developed at the University of Ghent by the Department of Electronics and Information Technology [36]. The structures of these PSC devices and the crystal structure of KSnI_3_ perovskite material are illustrated in Figure 1 and Figure 2, while the band alignment of ETLs and HTL with the perovskite layer is shown in Figure 3.

The simulations in the SCAPS-1D package are based on solving the following set of equations [29,36]:(1)$$\frac{\partial E}{\partial x}=\frac{\partial }{\partial x}\left(\varepsilon \frac{\partial \Psi }{\partial x}\right)=\frac{-q}{\varepsilon_{0}}\left[-n+p-N_{A}+N_{D}+p_{t}-n_{t}\right]$$(2)$$J_{n}=q\mu_{n}nE+qD_{n}\frac{\partial n}{\partial x}$$(3)$$J_{p}=q\mu_{p}pE+qD_{p}\frac{\partial p}{\partial x}$$(4)$$\frac{\partial n}{\partial t}=G_{n}-U_{n}\left(n,p\right)+\frac{\partial J_{n}}{\partial x}$$(5)$$\frac{\partial p}{\partial t}=G_{p}-U_{p}\left(n,p\right)+\frac{\partial J_{n}}{\partial x}$$

In Poison’s Equation (1), the quantities *E*, *ψ*, *ε*, *ε*_0_, *n*, *p*, *N_A_*, *N_D_*, *p_t_*, and *n_t_* are the electric field, electrostatic potential, relative permittivity, permittivity of free space, the concentration of electrons, concentration of holes, density of ionized acceptor dopants, the density of ionized donor dopants, trapped hole concentration, and electron concentration, respectively. In the transport Equations (2) and (3), the parameters *μ_n_*, *μ_p_*, *D_n_*, and *D_p_*, are electron mobility, hole mobility, electron diffusion coefficient, and hole diffusion coefficient, respectively, while in the continuity Equations (4) and (5), *G_n_*, *G_p_*, *U_n_*(*n*, *p*), and *U_p_*(*n*, *p*) are electron generation rate, hole generation rate, electron recombination rate, and hole recombination rate, respectively. Poisson’s equation describes the relationship between charge and electric field within a PSC, while the transport equations account for the migration of electrons and holes in the device, and the continuity equations trace the generation, recombination, and migration of electrons and holes in the PSC.

Table 1 presents the values of the properties of the KSnI_3_, ETL, HTL, and FTO used as input data in the SCAPS-1D simulation of the four perovskite solar cell devices. The parameters such as the thickness and dopant densities of different layers were later updated during the optimization process. Interface defect layers were also included using the data presented in Table 2.

In addition to the structural and electronic properties shown in Table 1 and Table 2, the simulation of perovskite solar cells requires the knowledge of the absorption coefficients of the active absorption layer material used in the simulations. In the SCAPS-1D simulation package, the absorption coefficients, *Φ*, are given by [28,29].(6)$$\Phi \left(\lambda \right)=\left(\alpha +\frac{\beta E_{g}}{hv}\right)+\sqrt[{}]{\frac{hv}{E_{g}}-1},$$
where the model parameters *α* and *β* are directly proportional to $\sqrt{E_{g}}$ and inversely proportional to $\sqrt{E_{g}}$, respectively, *E_g_* is the active absorption layer band gap, while *h* and *v* are Planck’s constant and the frequency of incident photons, respectively.

### 2.2. Model and Material Validation

It was necessary to conduct SCAPS-1D model validation before simulating the new perovskite solar cell structures. In this research, we benchmarked our simulations by reproducing the FTO/SnO_2_/3C-SiC/KSnI_3_/NiO/C perovskite solar cell structure, which was optimized and proposed by Ref. [27] and is currently the most efficient KSnI_3_-based perovskite solar configuration in the literature. Table 3 shows the structural and electronic properties used in our benchmark simulation.

The corresponding performance metrics, which were obtained from this research, are presented in Table 4 where they are compared with the results of Ref. [27]. Our simulations yielded a PCE of 22.31%, which is very close to the PCE of 22.78% obtained by Ref. [27], thus providing confidence in our SCAPS-1D model as well as the properties of FTO, SnO_2_, and KSnI_3_ that we incorporated in the SCAPS-1D software. Similarly, the accuracy of properties of Al–ZnO, LiTiO_2_, ZnO, and rGO, which were also used in this research, were verified in our previous study, as discussed in Ref. [29]. Thus, we have confidence in our SCAPS-1D model and the properties of all materials used in this study.

## 3. Results

The optimization strategy followed in this study is two-fold. We started by optimizing the FTO/Al–ZnO/KSnI_3_/rGO/Se structure by varying the thickness and dopant density of each layer in this structure. In this optimized structure, we substituted Al–ZnO with other ETLs, namely LiTiO_2_, ZnO, and SnO_2_, of which we also optimized the thickness and donor dopant density to obtain optimized FTO/LiTiO_2_/KSnI_3_/rGO/Se, FTO/ZnO/KSnI_3_/rGO/Se, and FTO/SnO_2_/KSnI_3_/rGO/Se. The optimization of the aforementioned parameters was carried out as a result of the reasons discussed below.

The main function of the HTL in a perovskite solar cell is to effectively prevent the migration of electrons from the perovskite layer (PL) to the HTL and allow the transmission of holes from the PL to the HTL and the anode [38]. However, if the thickness of the HTL is higher than the optimal thickness, the hole will have long paths to migrate within the HTL. As a result, the recombination probability will increase and the performance of the PSC will be poor. Similarly, if the HTL is thinner than its optimal thickness, the HTL will not be effective in extracting holes from the PL. Hence, the optimization of the HTL thickness is very crucial for improving the PCE of the perovskite solar cell. Similarly, the key function of the ETL is to extract electrons from the PL and transport them to the cathode. Its thickness also plays a vital role in the device’s performance. For instance, it may yield low efficiency in charge collection if it is too thin, while it may decrease conductivity if it is too thick [39]. Furthermore, the major roles of a perovskite layer in a PSC are the attenuation of the solar spectrum within the visible-light energy range and the generation of electron–hole pairs through the photovoltaic process. The thickness of the PL highly affects its performance. For instance, if the PL is too thin it will not have sufficient volume to absorb light, and thus yield a low density of excitons. On the other hand, if it is too thick, it reduces the performance of the PSC since the recombination rate increases with the increase in the PL thickness [30]. Therefore, it is very critical that the thickness of the PL is optimized to achieve the highest possible performance of the PSC.

Similarly, the acceptor dopant density of the HTL, donor dopant density of the ETL, and acceptor dopant density of the active perovskite absorber layer play very crucial roles in the performance of perovskite solar cells. In particular, the separation of electron–hole pairs in PSC devices improves with the enhancement in the electric field that exists at the interface between the ETL and absorber layer and at the interface between the HTL and absorber layer. The electric field strength increases with the increase in the donor dopant density of the ETL and acceptor dopant density of the HTL, and this improves the performance of the device [8]. Even the electric field within the PL depends on the dopant density of the PL, and this electric field is also responsible for the reduction in recombinations within the PL [8]. However, if the dopant density is too high, it results in Coulomb traps that increase the likelihood of charge recombination. Thus, finding the optimal dopant densities of the ETL, HTL, and PL is very crucial for every PSC device.

### 3.1. Optimization of the Hole Transport Layer Thickness

In this research, we optimized the thickness of rGO HTL, by varying it from 20 nm to 4000 nm, and assessed its impact on the open circuit voltage (V_oc_), short-circuit current (J_sc_), fill-factor (FF), and power conversion efficiency (PCE). The rest of the parameters were kept constant at their initial values shown in Table 1. The results are depicted in Figure 4. We observe that the J_sc_ increases rapidly as the rGO thickness increases towards 1000 nm, and thereafter remains relatively constant. On the other hand, V_oc_ is relatively constant. In particular, J_sc_ ranges from 11.823 to 22.182 mA/cm^2^, while V_oc_ ranges from 1.334 to 1.361 V. On the other hand, FF somewhat decreases as the rGO thickness approaches 500 nm, beyond which it is practically constant. The PCE shows a similar trend as J_sc_. Quantitatively, FF ranges between 87.99% and 87.39%, while PCE is in the range of 13.88 to 26.38%. The trends observed on the performance metrics are similar to the recent results of Refs. [8,20,40]. The increase in the V_oc_ and PCE is attributed to the enhancement in hole extraction from the perovskite layer and transportation to the anode as the HTL thickness increases. The reduction in FF with the increase in HTL thickness may be due to the lateral resistance in the junction between HTL and the active absorption layer. The highest PCE is observed at 2670 nm. Thus, 2670 nm was deemed the optimal thickness of the rGO layer.

### 3.2. Optimization of the Hole Transport Layer Dopant Density

In this study, the acceptor dopant density of rGO was optimized by varying it between 2.5 × 10^18^ cm^−3^ and 10^22^ cm^−3^, and the dependence of V_oc_, J_sc_, FF, and PCE on HTL acceptor dopant density was examined, as shown in Figure 5. The thickness of this HTL was fixed at its optimal value of 2670 nm obtained in Section 3.1, while the rest of the parameters were fixed at their values shown in Table 1. The results show that J_sc_ and V_oc_ do not change with the change in the acceptor dopant density of the rGO HTL. In contrast, FF and PCE somewhat increase as the acceptor dopant density rises towards 10^19^ cm^−3^ and 10^20^ cm^−3^, respectively, and thereafter remain constant. The notable increase in FF and PCE is similar to other observations in the literature [41], and is due to the enhancement in the exciton separation as the result of the increase in electric field strength at the HTL/absorber interface [42]. In particular, J_sc_ is in the range of 22.17 to 22.26 mA/cm^2^, and V_oc_ is 1.36 V. On the other hand, FF and PCE are in the ranges of 86.51 to 87.38% and 25.93 to 26.38%, respectively. This highest PCE of 26.38% is observed at the acceptor dopant density of 5.45 × 10^21^ cm^−3^. As a result, 5.45 × 10^21^ cm^−3^ was considered the optimal acceptor dopant density of rGO.

### 3.3. Optimization of Perovskite Layer Thickness

In the present study, we optimized the thickness of the KSnI_3_ active absorption layer, by varying it from 50 nm to 1000 nm, and kept the thickness and acceptor dopant density of the HTL at their optimal values obtained in Section 3.1 and Section 3.2, while the rest of the parameters were kept at their respective values provided in Table 1. The variation of V_oc_, J_sc_, FF, and PCE with PL thickness is depicted in Figure 6. We observe that J_sc_ and V_oc_ do not significantly change as the KSnI_3_ layer thickness increases. On the other hand, the FF and PCE are decreasing functions of PL thickness in the entire 50 nm to 1000 nm region. In particular, J_sc_, V_oc_, FF, and PCE range from 22.02 mA/cm^2^ to 22.18 mA/cm^2^, 1.34 V to 1.36 V, 87.65% to 83.75%, and 26.50% to 24.76%, respectively. Clearly, the performance of the PL deteriorates with the increase in its thickness. This is due to the increase in recombination rate with the increase in the perovskite layer thickness [30,39]. Thus, the optimal thickness of the PL was considered to be 50 nm, at which the highest PCE is observed.

### 3.4. Optimization of Perovskite Layer Acceptor Dopant Density

We optimized the acceptor dopant density of the perovskite layer by varying it from 10^19^ cm^−3^ to 10^22^ cm^−3^ and examined the dependence of the V_oc_, J_sc_, FF, and PCE on the dopant density. During this process, the thickness and dopant density of the HTL were at their optimal values obtained in Section 3.1 and Section 3.2, the thickness of the PL was fixed at its optimal value obtained in Section 3.3, and the rest of the parameters were kept at their respective values provided in Table 1. Figure 7 presents V_oc_, J_sc_, FF, and PCE as a function of the acceptor dopant density of the PL. It was observed that J_sc_ decreases with the increase in dopant density, and it ranges between 14.40 mA/cm^2^ and 22.16 mA/cm^2^. In contrast, V_oc_ increases slowly with the increase in the PL acceptor dopant density. In particular, it ranges from 1.36 V to 1.48 V. This rise in V_oc_ is the reflection of the increase in built-in potential, resulting from the increase in the acceptor dopant density [43]. Similarly, the FF rises as the dopant density increases towards 10^21^ cm^−3^ and, thereafter, remains relatively constant. In particular, it is in the range of 84.10% to 91.10%. This increase in the FF corresponds to the reduction in the resistivity of the absorption layer [44]. On the other hand, the PCE slightly rises as the dopant density approaches 10^21^ cm^−3^, above which it is a decreasing function of the acceptor dopant density. Quantitatively, PCE has the lowest value of 19.45% and the highest value of 27.36%. The notable reduction in J_sc_ and PCE with the increase in dopant density may be attributed to the decrease in charge carrier mobility and rise in charge recombination rate [16,45,46]. The highest PCE occurs at the dopant density of 8.33 × 10^20^ cm^−3^, which is, therefore, considered the optimal acceptor dopant density of the PL.

### 3.5. Optimization of Electron Transport Layer Thickness

In the present study, the thickness of the Al–ZnO electron transport layer was optimized through its variation from 50 nm to 4000 nm. The other parameters were fixed at their optimal values found in Section 3.1, Section 3.2, Section 3.3 and Section 3.4, while those that were not optimized were kept at their initial values shown in Table 1. The dependence of V_oc_, J_sc_, FF, and PCE on the Al–ZnO thickness is presented in Figure 8. The J_sc_ and PCE are relatively constant at thicknesses smaller than 1000 nm, and then slowly decline as the thickness increases. In particular, J_sc_ is in the range of 20.56 to 21.16 mA/cm^2^, while PCE ranges between 26.58% and 27.37%. The V_oc_ and FF, on the other hand, remain practically stable at 1.43 V and 90.52% in the entire 50 nm to 4000 nm ETL thickness range. Thus, it is clear that the overall performance of the PSC device slightly declines as the Al–ZnO becomes thicker. This observation is consistent with the literature. It could be attributed to the increase in resistance and deterioration in conductivity as a result of the increase in ETL thickness [29,39]. The highest PCE is observed at the Al–ZnO thickness of 489 nm, which was, thus, considered the optimal thickness of the Al–ZnO ETL.

### 3.6. Optimization of Electron Transport Layer Donor Dopant Density

In this research, we found the optimal donor dopant density of the Al–ZnO electron transport layer by examining how it affects V_oc_, J_sc_, FF, and PCE. These performance metrics were assessed in the 10^15^ to 10^20^ cm^−3^ donor dopant density range, as depicted in Figure 9. The results reveal that J_sc_ and V_oc_ are unaffected by the change in donor dopant density of Al–ZnO, and they are in the range of 21.16 to 21.17 mA/cm^2^ and 1.41 to 1.43 V, respectively. The FF and PCE are relatively constant at donor dopant densities smaller than 10^18^ cm^−3^, and thereafter slowly decrease with the increase in the donor dopant density. In particular, they are in the range of 89.69 to 90.81% and 27.50 to 26.86%, respectively. Thus, the overall performance of the device slightly deteriorates with the increase in the donor dopant density of the Al–ZnO layer, at high dopant densities. This may be due to Coulomb traps, which are understood to result from high dopant concentrations and cause a reduction in charge mobility [27,47]. Since PCE is relatively constant up to 10^18^ cm^−3^, the dopant density of 10^15^ cm^−3^ was deemed the optimal donor dopant density of the Al–ZnO ETL.

### 3.7. Optimization of FTO Layer Thickness

We also optimized the thickness of the FTO layer, by varying it between 100 nm and 4000 nm, and assessed the impact of this on the V_oc_, J_sc_, FF, and PCE, as presented in Figure 10. The results show that the performance metrics do not change much as the thickness increases. The J_sc_ varies between 20.91 mA/cm^2^ and 21.28 mA/cm^2^, and V_oc_ remains at 1.43 V. FF ranges from 90.60 to 90.68%, and PCE ranges from 27.15 to 27.60%. Thus, the device performance is not highly affected by the FTO thickness. The slight decline in J_sc_ and PCE at low FTO thickness may be attributed to the increase in series resistance and optical losses, which are correlated to the increase in the FTO thickness. The highest PCE occurs at a thickness of 100 nm, which was, therefore, deemed the optimal thickness of the FTO layer.

This is the end of the optimization of the FTO/Al–ZnO/KSnI_3_/rGO/Se structure. It was used as the framework for optimization of the FTO/LiTiO_2_/KSnI_3_/rGO/Se, FTO/ZnO/KSnI_3_/rGO/Se, and FTO/SnO_2_/KSnI_3_/rGO/Se configurations, as presented in the next sections.

### 3.8. Substitution of Al–ZnO Electron Transport Layer

In further analysis, we found the optimal FTO/LiTiO_2_/KSnI_3_/rGO/Se, FTO/ZnO/KSnI_3_/rGO/Se, and FTO/SnO_2_/KSnI_3_/rGO/Se PSC configurations by substituting Al–ZnO with LiTiO_2_, ZnO, and SnO_2_ in the optimized FTO/Al–ZnO/KSnI_3_/rGO/Se structure. During this process, the thicknesses and dopant densities of the rGO, KSnI_3_, and FTO layers were kept at their optimal values found in the previous sections, while the thicknesses and donor dopant densities of LiTiO_2_, ZnO, and SnO_2_ were varied to assess their impact on V_oc_, J_sc_, FF, and PCE.

#### 3.8.1. Optimization of LiTiO_2_, ZnO, and SnO_2_ Thicknesses

The ETL thicknesses of the FTO/LiTiO_2_/KSnI_3_/rGO/Se, FTO/ZnO/KSnI_3_/rGO/Se, and FTO/SnO_2_/KSnI_3_/rGO/Se perovskite solar cell structures were optimized by varying the thicknesses of LiTiO_2_, ZnO, and SnO_2_ from 100 nm to 4000 nm, and we observed the dependence of V_oc_, J_sc_, FF, and PCE on the thickness. The results are depicted in Figure 11. We observe that for ETL = LiTiO_2_, ZnO, and SnO_2_, V_oc_ is relatively constant in the entire 100 to 4000 nm ETL thickness range. In particular, V_oc_ is 1.30 V, 1.43 V, and 1.53 V for LiTiO_2_, ZnO, and SnO_2_, respectively. The J_sc_ corresponding to SnO_2_ and ZnO is relatively stable in the whole thicknesses range, while for LiTiO_2_ it slightly decreases as the thickness approaches 1500 nm and thereafter remains constant. In particular, J_sc_ is in the range of 20.13 to 21.25 mA/cm^2^, 20.93 to 21.26 mA/cm^2^, and 21.16 to 21.25 mA/cm^2^ for LiTiO_2_, ZnO, and SnO_2_, respectively.

The results also show that FF is practically constant for all ETL materials. Quantitatively, it is in the range of 91.25 to 91.27% for SnO_2_, 90.04 to 90.81% for ZnO, and 89.72 to 89.95% for LiTiO_2_. Furthermore, PCE shows trends similar to J_sc_. It is relatively stable for ZnO and SnO_2_, while for LiTiO_2_ it somewhat decreases as the thickness rises towards 1000 nm and thereafter remains unaffected by the thickness. In particular, it ranges from 23.57 to 24.91% for LiTiO_2_, 27.03 to 27.64% for ZnO, and 29.53 to 29.67% for SnO_2_. Thus, the thicknesses of ZnO and SnO_2_ do not have a significant impact on the overall performance of FTO/ZnO/KSnI_3_/rGO/Se and FTO/SnO_2_/KSnI_3_/rGO/Se devices, respectively, while the increase in the thickness of LiTiO_2_ slightly reduces the overall performance of the FTO/LiTiO_2_/KSnI_3_/rGO/Se device. The deterioration in the PCE and J_sc_ for LiTiO with the increase in the ETL thickness may result from an increase in series resistance and the low transmission of light through the ETL layer [44]. The highest PCE is observed at 50 nm for LiTiO_2_ and ZnO, and 489 nm for SnO_2_. Thus, these were considered the optimal thicknesses of LiTiO_2_, ZnO, and SnO_2_, in the FTO/LiTiO_2_/KSnI_3_/rGO/Se, FTO/ZnO/KSnI_3_/rGO/Se, and FTO/SnO_2_/KSnI_3_/rGO/Se PSC structures, respectively.

#### 3.8.2. Optimization of LiTiO_2_, ZnO, and SnO_2_ Donor Dopant Densities

The donor dopant densities of the LiTiO_2_, ZnO, and SnO_2_ electron transport layers were optimized, by varying them between 10^12^ and 10^19^ cm^−3^, and observing their effect on the J_sc_, V_oc_, FF, and PCE, while keeping the rest of the parameters at their optimal values found in the previous sections. As depicted in Figure 12, the results show that J_sc_, V_oc_, FF, and PCE are not significantly affected by the donor dopant densities of LiTiO_2_, ZnO, and SnO_2_. Thus, the donor dopant densities of LiTiO_2_, ZnO, and SnO_2_ do not have a significant impact on the overall performance of the FTO/LiTiO_2_/KSnI_3_/rGO/Se, FTO/ZnO/KSnI_3_/rGO/Se, and FTO/SnO_2_/KSnI_3_/rGO/Se solar cells, respectively. These observations are consistent with the recent literature [20]. In particular, J_sc_ is in the range of 21.25 to 21.26 mA/cm^2^ for LiTiO_2_, 21.25 to 21.26 mA/cm^2^ for SnO_2_, and 21.26 to 21.27 mA/cm^2^ for ZnO. V_oc_ is 1.30 V for LiTiO_2_, 1.53 V for SnO_2_, and 1.43 V for ZnO. Furthermore, FF is 89.63 to 90.01% for LiTiO_2_, 91.01 to 91.28% for SnO_2_, and 90.52 to 90.79% for ZnO. The PCE, on the other hand, is in the range of 24.77 to 24.94% for LiTiO_2_, 29.67 to 29.53% for SnO_2_, and 27.51 to 27.62% for ZnO. The highest PCEs are observed at the donor dopant densities of 10^12^ cm^−3^ for LiTiO_2_ and 10^16^ cm^−3^ for SnO_2_ and ZnO. Thus, these were deemed the optimal donor dopant densities for ETL = LiTiO_2_, ZnO, and SnO_2_.

### 3.9. Optimized Parameters and Performance Metrics

Table 5 presents the optimized parameters of FTO/Al–ZnO/KSnI_3_/rGO/Se, FTO/LiTiO_2_/KSnI_3_/rGO/Se, FTO/ZnO/KSnI_3_/rGO/Se, and FTO/SnO_2_/KSnI_3_/rGO/Se PSC structures. The rest of the parameters remained unchanged, as shown in Table 1.

Table 6 presents the performance metrics of the optimized PSC structures. These correspond to the optimized structural and electronic properties shown in Table 5. It is clear that FTO/SnO_2_/KSnI_3_/rGO/Se has the highest PCE. Thus, it became our main focus in the rest of the analysis.

### 3.10. Effect of Series and Shunt Resistances

Series resistance (R_s_) and shunt resistance (R_sh_) are some of the crucial parameters that highly affect the performance of perovskite solar cells. Failure to optimize them may result in poor performance of the PSC device. Series resistance is understood to exist at various contact points of the device such as anode, cathode, and interfaces. On the other hand, shunt resistance is a desired property in a PSC, and it prevents the flow of undesired shunt current that reduces the net current of the device. In this study, we investigated the effect of the R_s_ and R_sh_ on the performance of our highest-performing PSC structure, which is FTO/SnO_2_/KSnI_3_/rGO/Se. In particular, we varied R_s_ and R_sh_ from 0 to 8 Ω cm^2^ and 0 to 10^8^ Ω cm^2^, respectively, while other parameters were kept at the optimal values found in the previous sections. The results are presented in Figure 13 and Figure 14. We observe that J_sc_ and V_oc_ are unaffected by the change in R_s_, while FF and PCE are both decreasing functions of R_s_. In particular, J_sc_ and V_oc_ are 21.25 mA/cm^2^ and 1.53 V, respectively, while FF and PCE are in the range of 91.27 to 80.51% and 29.67 to 26.19%, respectively. Thus, the overall performance of the FTO/SnO_2_/KSnI_3_/rGO/Se device deteriorates with the increase in R_s_. These results are similar to other observations in the literature, e.g., Ref. [30]. The decline in the FF with an increase in R_s_ is consistent with the dependence of FF on R_s_, given by FF = FF_0_(1 − R_s_), where FF_0_ is the fill factor in the absence of series resistance [28].

Furthermore, the change in R_sh_ also does not affect the J_sc_, while V_oc_, FF, and PCE rapidly rise as the R_sh_ approaches 10^7^ Ω cm^2^, above which they remain relatively stable. This is consistent with the observations of Ref. [28]. Quantitatively, J_sc_ is 21.25 mA/cm^2^ in the whole R_sh_ region, while V_oc_, FF, and PCE are in the range of 0.213 to 1.53 V, 18.74 to 91.27%, and 0.8463 to 29.67%, respectively. Thus, 10^7^ Ω cm^2^ was deemed the optimal R_sh_ for the FTO/SnO_2_/KSnI_3_/rGO/Se PSC device.

### 3.11. Effect of Temperature

Temperature is also a critical factor that can highly influence the overall performance of the perovskite solar device. In this study, we examined the effect of temperature on J_sc_, V_oc_, FF, and PCE of the FTO/SnO_2_/KSnI_3_/rGO/Se configuration by varying the temperature from 300 K to 400 K. The rest of the parameters were fixed at their optimal values found above.

Figure 15 shows the variations of V_oc_, J_sc_, FF, and PCE with temperature and reveals that V_oc_ remains constant, while J_sc_ slowly rises with the increase in temperature. The gradual increase in J_sc_ corresponds to the reduction in the band gap of the absorption layer, resulting in the enhancement in the generation of electron–hole pairs [48,49]. The PCE follows the same increasing trend as J_sc_. This is similar to the recent observations of Ref. [50]. The slight enhancement in the PCE is attributed to the improved charge-carrier mobility and exciton generation, which are understood to increase with increases in the device temperature until the optimal temperature is reached where non-radiative recombinations and resistance in transport layers start dominating [51]. On the other hand, FF decreases with the increase in temperature. This is consistent with the temperature dependence of FF given by [6].(7)$$FF=\frac{\frac{qV_{oc}}{nkT}-ln\left(\frac{qV_{oc}}{nkT}+0.72\right)}{\frac{qV_{oc}}{nkT}+1}$$
where *nkT/q* is the thermal voltage. Quantitatively, J_sc_, *V_oc_*, *FF*, and PCE range from 21.25 to 22.15 mA/cm^2^, 1.53 to 1.54 V, 91.27 to 89.20%, and 29.67 to 30.44%, respectively. The highest PCE is observed at 371 K, and this was considered the optimal operation temperature for the FTO/SnO_2_/KSnI_3_/rGO/Se PSC structure.

### 3.12. Defect Density of the Perovskite Layer

The possible poor quality of the of a perovskite active absorption layer may result in the presence of defects in the active layer. These defects can highly affect the performance of a perovskite solar cell. In particular, they become trap sites where Shockley–Read–Hall (SRH) recombinations take place. Thus, in addition to the parameters investigated above, we also investigated the impact of defect density of the perovskite layer in our highest-performing FTO/SnO_2_/KSnI_3_/rGO/Se structure, by varying the defect density while keeping the rest of the parameters at their optimized values. The results are depicted in Figure 16 and show that the overall performance of the device remains relatively stable as the total defect density increases from 1013 to 1015 cm^−3^, beyond which the overall performance starts deteriorating. This means that 1015 cm^−3^ is the highest and worse possible defect density that would still yield no significant decrease in the PCE. Thus, the defect density of 1015 cm^−3^ was deemed the optimal defect density of the KSnI_3_ in the FTO/SnO_2_/KSnI_3_/rGO/Se structure, and it corresponds to V_oc_, J_sc_, FF, and PCE of 1.54 V, 22.04 mA/cm^2^, 89.76%, and 30.44%, respectively.

Table 7 presents the summary of the performance metrics of the optimized PSC structures when the optimal R_s_, R_sh_, and temperature of 0, 10^7^ Ω cm^2^, and 371 K, respectively, are assumed. These also correspond to the optimized structural and electronic properties shown in Table 5 and the perovskite layer defect density of 10^15^ cm^−3^.

### 3.13. Comparison of Results with Literature

Table 8 shows the V_oc_, J_sc_, FF, and PCE of the optimized FTO/Al–ZnO/KSnI_3_/rGO/Se, FTO/LiTiO_2_/KSnI_3_/rGO/Se, FTO/ZnO/KSnI_3_/rGO/Se, and FTO/SnO_2_/KSnI_3_/rGO/Se PSC structures, and V_oc_, J_sc_, FF, and PCE of all KSnI_3_ perovskite solar cells available in the literature. All our investigated PSC configurations are higher than the PCE of 22.78%, which is currently the highest PCE of KSnI_3_-based perovskite solar cells. In fact, the highest performing PSC structure, FTO/SnO_2_/KSnI_3_/rGO/Se, has a PCE of 30.44% under ideal R_s_, R_sh_, and temperature conditions. This is 7.66% more efficient than the FTO/SnO_2_/3C–SiC/KSnI_3_/NiO/C, which is the highest-performing PSC known in the literature thus far. However, in reality there is no perovskite solar cell that has completely zero series resistance. Hence, we also simulated our highest-performing configuration, FTO/SnO_2_/KSnI_3_/rGO/Se, using the R_s_ = 0.5 Ω cm^2^ suggested by Ref. [28], and obtained performance metrics shown in Table 9. The results show that the performance does not change significantly when there is a small non-zero series resistance, and the device yields a PCE of 30.21%.

Although SCAPS-1D has, in the past studies, reasonably reproduced experimental values of V_oc_, J_sc_, FF, and PCE for another Sn-based perovskite solar cell structure ITO/TiO_2_/CsSnBr_3_/Spiro-OMeTAD/Au [37], we cannot rule out its limitations discussed in detailed by Ref. [53], such as (i) the inability to account for reflection losses and (ii) its reliance on the theoretical optical absorption model for materials, such as KSnI_3_, for which there is no experimental data for optical absorption coefficients. Thus, the PCE results presented in Table 8 and Table 9 are predictions and do need experimental verifications in future studies.

Furthermore, Sn-based perovskite solar cells suffer from oxidation, which negatively affects their stability and performance. However, research has shown that there are various ways in which we can suppress oxidation [54,55]. For instance, (i) incorporating graphene tin quantum dot composites in a perovskite layer greatly improves stability and efficiency of the perovskite solar cell [54], and (ii) introducing additives such as tin fluoride in the active layer can also significantly reduce oxidation, and this approach is widely used and effective [55]. Thus, these methods can also be used in future experimental studies of our highest-performing FTO/SnO_2_/KSnI_3_/rGO/Se device in an attempt to eliminate oxidation, since it is also Sn based.

## 4. Conclusions

In this study, we optimized FTO/Al–ZnO/KSnI_3_/rGO/Se, FTO/LiTiO_2_/KSnI_3_/rGO/Se, FTO/ZnO/KSnI_3_/rGO/Se, and FTO/SnO_2_/KSnI_3_/rGO/Se PSC structures using the SCAPS-1D simulation package. In particular, we optimized the thicknesses and dopant densities of rGO, KSnI_3_, Al–ZnO, LiTiO_2_, ZnO, and SnO_2_ layers, the thickness of FTO, as well as the defect density of KSnI_3_. This yielded PCEs of 27.60%, 24.94%, 27.62%, and 30.21% for FTO/Al–ZnO/KSnI_3_/rGO/Se, FTO/LiTiO_2_/KSnI_3_/rGO/Se, FTO/ZnO/KSnI_3_/rGO/Se, and FTO/SnO_2_/KSnI_3_/rGO/Se, respectively. Thus, the PCE of FTO/SnO_2_/KSnI_3_/rGO/Se is 7.43% higher than the PCE of FTO/SnO_2_/3C–SiC/KSnI_3_/NiO/C, which is currently the highest performing KSnI_3_-based perovskite solar cell in the literature. Thus, we propose FTO/SnO_2_/KSnI_3_/rGO/Se as the new highest-performing KSnI_3_-based PSC. We also call for experimental studies to further verify and improve the proposed configuration. The ideal performance conditions of this newly developed PSC configuration are a series resistance of 0.5 Ω cm^2^, a shunt resistance of 10^7^ Ω cm^2^, and a temperature of 371 K.

## Figures and Tables

**Figure 1 nanomaterials-15-00580-f001:**
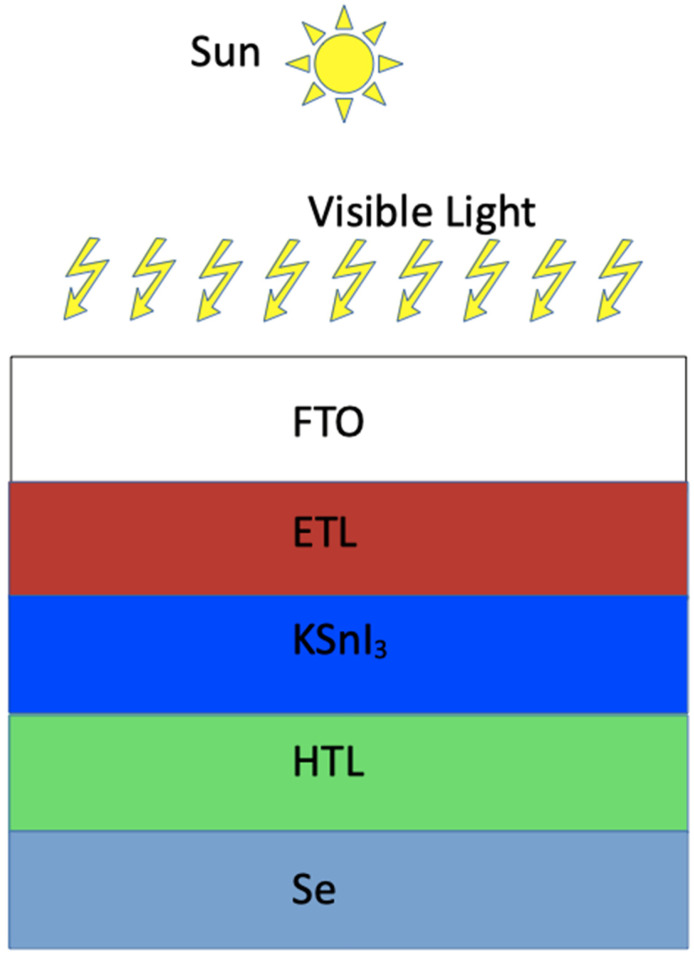
Schematic layout of FTO/ETL/KSnI_3_/HTL/Se device.

**Figure 2 nanomaterials-15-00580-f002:**
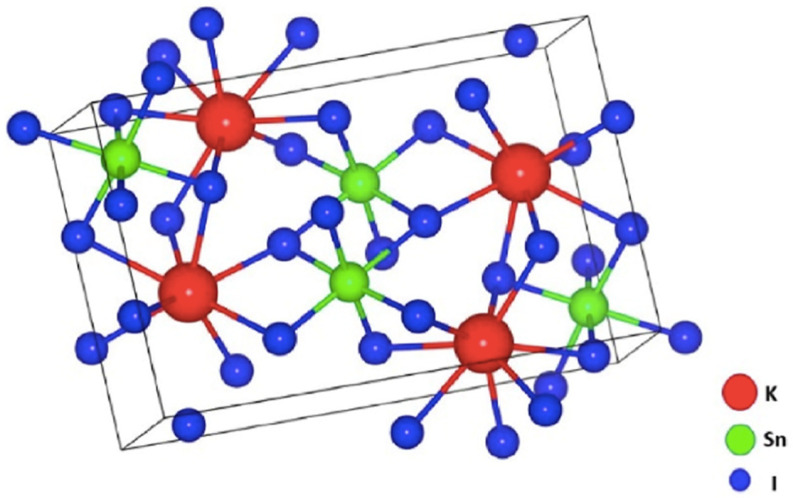
Structure of the active absorption layer material KSnI_3_ [22].

**Figure 3 nanomaterials-15-00580-f003:**
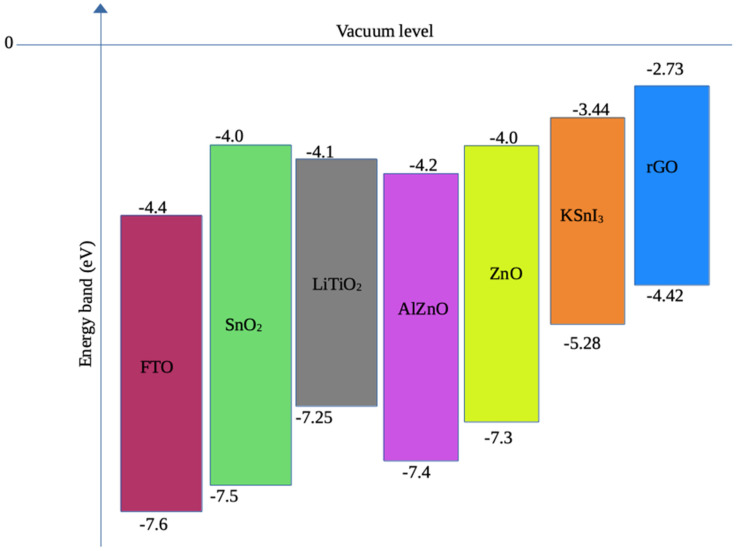
Valence band maxima and conduction band minima of FTO, ETLs, HTL, and KSnI_3_.

**Figure 4 nanomaterials-15-00580-f004:**
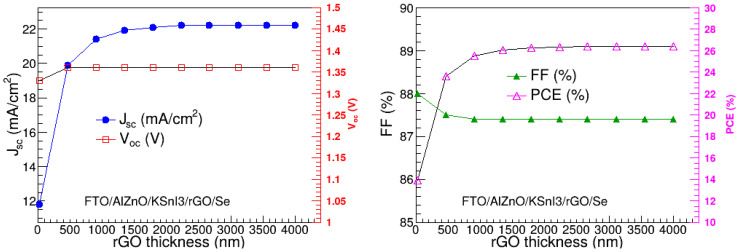
Dependence of V_oc_, J_sc_, FF, and PCE on rGO thickness.

**Figure 5 nanomaterials-15-00580-f005:**
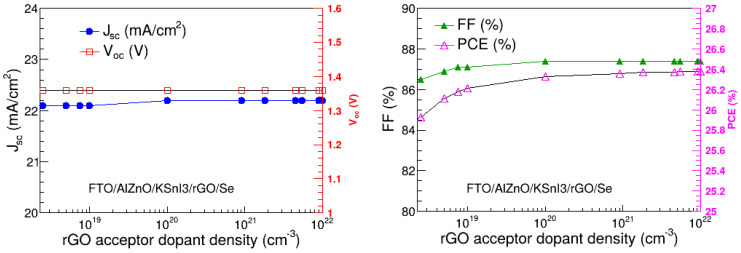
V_oc_, J_sc_, FF, and PCE as a function of rGO dopant density.

**Figure 6 nanomaterials-15-00580-f006:**
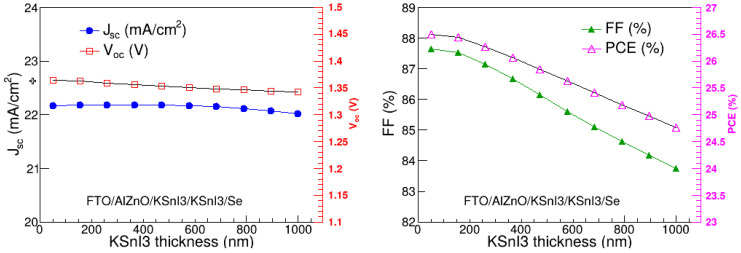
V_oc_, J_sc_, FF, and PCE as a function of PL thickness.

**Figure 7 nanomaterials-15-00580-f007:**
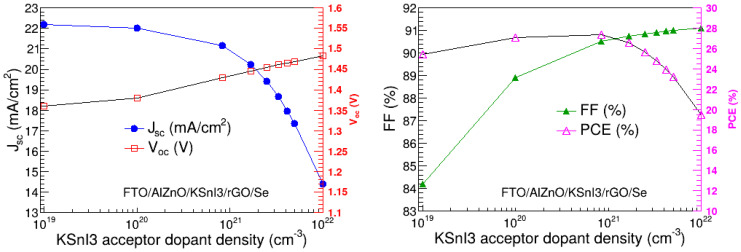
V_oc_, J_sc_, FF, and PCE as a function of PL acceptor dopant density.

**Figure 8 nanomaterials-15-00580-f008:**
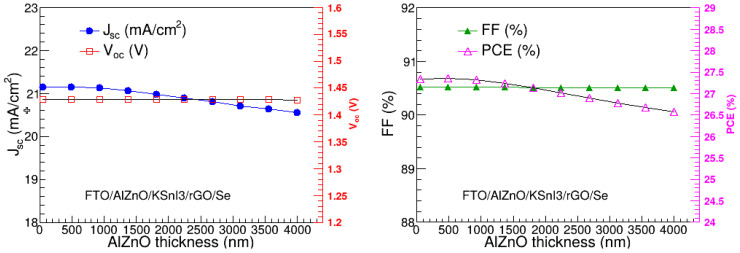
V_oc_, J_sc_, FF, and PCE as a function of Al–ZnO thickness.

**Figure 9 nanomaterials-15-00580-f009:**
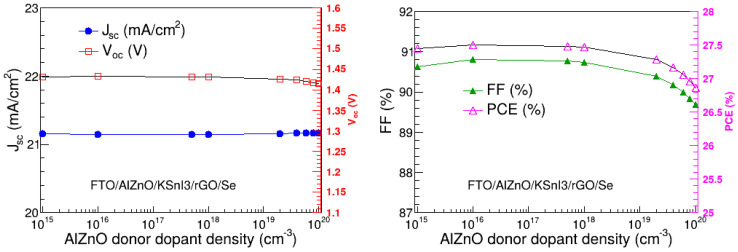
V_oc_, J_sc_, FF, and PCE as a function of Al–ZnO dopant density.

**Figure 10 nanomaterials-15-00580-f010:**
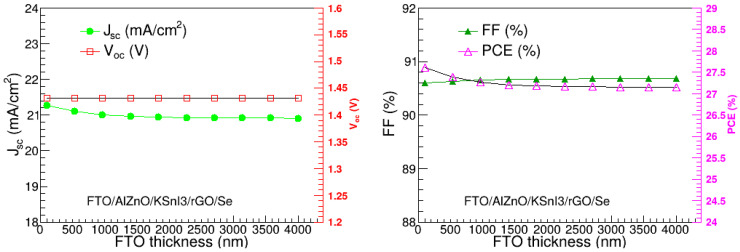
V_oc_, J_sc_, FF, and PCE as a function of FTO thickness.

**Figure 11 nanomaterials-15-00580-f011:**
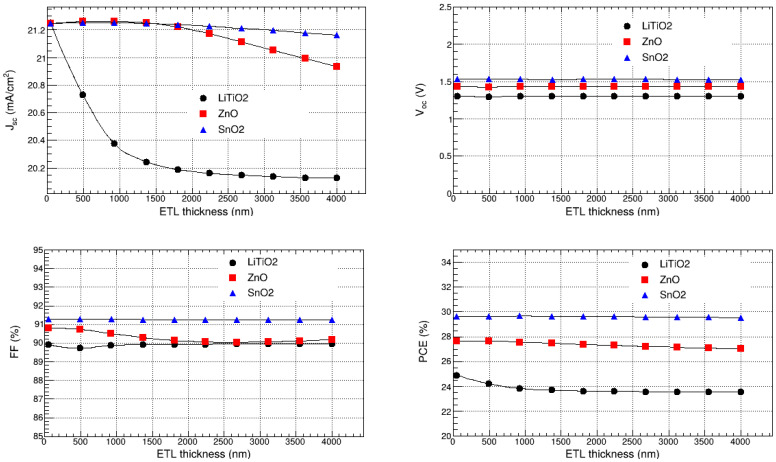
V_oc_, J_sc_, FF, and PCE as a function of the thicknesses of LiTiO_2_, ZnO, and SnO_2_.

**Figure 12 nanomaterials-15-00580-f012:**
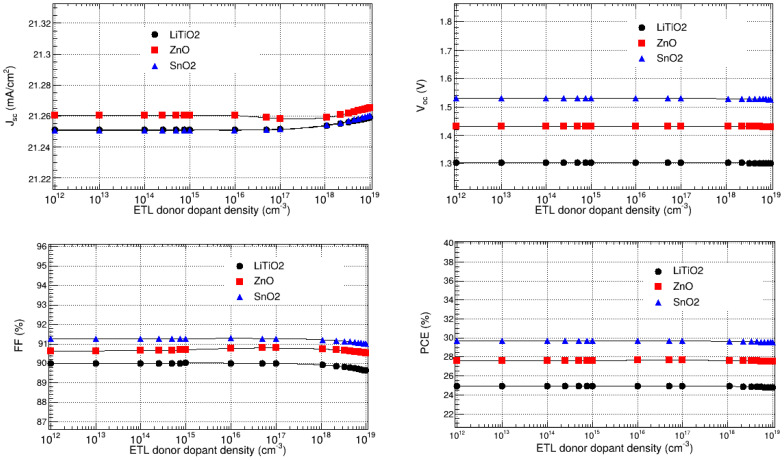
V_oc_, J_sc_, FF, and PCE as a function of the dopant densities of LiTiO_2_, ZnO, and SnO_2_.

**Figure 13 nanomaterials-15-00580-f013:**
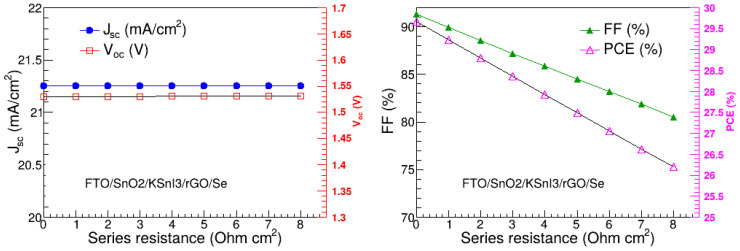
V_oc_, J_sc_, FF, and PCE as a function of series resistance.

**Figure 14 nanomaterials-15-00580-f014:**
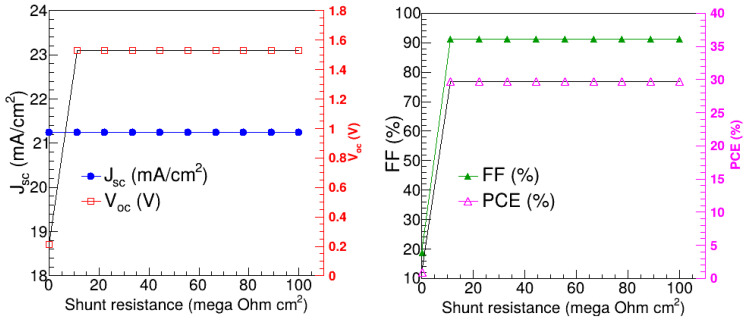
V_oc_, J_sc_, FF, and PCE as a function of shunt resistance for FTO/SnO_2_/KSnI_3_/rGO/Se.

**Figure 15 nanomaterials-15-00580-f015:**
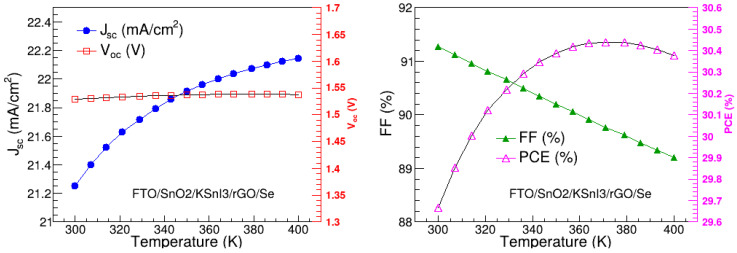
V_oc_, J_sc_, FF, and PCE as a function of temperature.

**Figure 16 nanomaterials-15-00580-f016:**
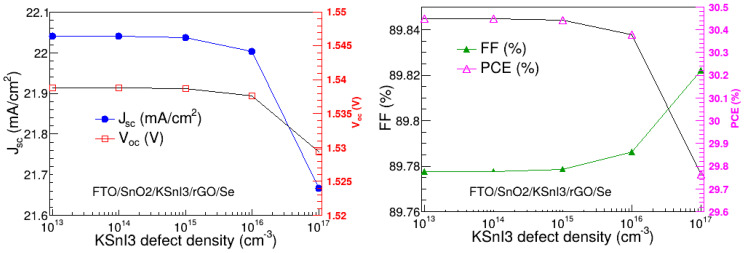
V_oc_, J_sc_, FF, and PCE as a function of KSnI_3_ defect density.

**Table 1 nanomaterials-15-00580-t001:** Properties of the KSnI_3_, ETL, HTL, and FTO used in the simulations before optimization [27,28,29].

Parameters	FTO	Al–ZnO	LiTiO_2_	ZnO	SnO_2_	KSnI_3_	rGO
Thickness (nm)	400	400	400	400	400	400	400
E_g_ (eV)	3.5	3.1	3.15	3.28	3.5	1.84 ^1^	1.69
χ (eV)	4	4	4	4	3.9	3.44 ^2^	3.56
ε_r_	9	9	13.6	9	9	10.4	13.3
N_c_ (cm^−3^)	2.02 × 10^18^	2 × 10^18^	3 × 10^20^	2 × 10^18^	2.2 × 10^18^	2.2 × 10^18^	10^18^
N_v_ (cm^−3^)	1.8 × 10^19^	1.8 × 10^19^	2 × 10^20^	1.8 × 10^19^	1.8 × 10^19^	1.8 × 10^19^	1.8 × 10^19^
V_th,n_ (cm s^−1^)	10^7^	10^7^	10^7^	10^7^	10^7^	10^7^	10^7^
V_th,h_ (cm s^−1^)	10^7^	10^7^	10^7^	10^7^	10^7^	10^7^	10^7^
μ_n_ (cm^2^/V/s)	2 × 10^1^	13.84	30	43	200	21.28	2.6 × 10^1^
μ_p_ (cm^2^/V/s)	1 × 10^−1^	25	0.01	25	80	19.46	1.23 × 10^2^
N_A_ (cm^−3^)	0	0	0	0	0	1 × 10^16^	10^22^
N_D_ (cm^−3^)	2 × 10^19^	1.02 × 10^19^	2 × 10^17^	2.9 × 10^15^	1 × 10^17^	1 × 10^15^	0
N_t_ (cm^−3^)	10^15^	10^15^	10^14^	10^15^	10^15^	10^15^	10^14^

^1^ Obtained from DFT calculations of Ref. [22] where the generalized gradient approximation was used. ^2^ Calculated as the negative value of the conduction band minimum, as shown in the DFT study of Ref. [22].

**Table 2 nanomaterials-15-00580-t002:** Interface defect input parameters [27,28].

Interface	HTL/KSnI_3_	ETL/KSnI_3_
Defect type	Neutral	Neutral
Energetic distribution	Single	Single
Capture cross-section for electrons (cm^−2^)	10^−19^	10^−19^
Capture cross-section for holes (cm^−2^)	10^−19^	10^−19^
Reference for defect energy level	Above VB maximum	Above VB maximum
Energy with respect to reference (eV)	0.6	0.6
Total density (cm^−3^)	1 × 10^10^	1 × 10^10^

**Table 3 nanomaterials-15-00580-t003:** Properties used in the benchmark simulation of FTO/SnO_2_/3C-SiC/KSnI_3_/NiO/C PSC [27].

Parameters	FTO	SnO_2_	3C-SiC	KSnI_3_	NiO
Thickness (nm)	500	25	20	1500	20
E_g_ (eV)	3.2	3.5	2.36	1.84	3.8
χ (eV)	4	3.9	3.8	3.44	1.4
ε_r_	9	9	9.72	10.4	10.7
N_c_ (cm^−3^)	2.2 × 10^18^	2.2 × 10^18^	1.553 × 10^19^	2.2 × 10^18^	2.5 × 10^19^
N_v_ (cm^−3^)	1.8 × 10^18^	1.8 × 10^19^	1.163 × 10^19^	1.8 × 10^19^	2.8 × 10^19^
V_th,n_ (cm s^−1^)	10^7^	10^7^	10^7^	10^7^	10^7^
V_th,h_ (cm s^−1^)	10^7^	10^7^	10^7^	10^7^	10^7^
μ_n_ (cm^2^/V/s)	2 × 10^1^	200	900	21.28	12
μ_p_ (cm^2^/V/s)	1 × 10^1^	80	40	19.46	28
N_A_ (cm^−3^)	0	0	0	1 × 10^16^	1 × 10^19^
N_D_ (cm^−3^)	2 × 10^19^	1 × 10^20^	1 × 10^20^	1 × 10^15^	0
N_t_ (cm^−3^)	10^14^	10^15^	10^14^	10^14^	10^15^

**Table 4 nanomaterials-15-00580-t004:** Benchmark results for the FTO/SnO_2_/3C-SiC/KSnI_3_/NiO/C configuration.

PSC Structure	V_oc_ (V)	J_sc_ (mA/cm^2^)	FF (%)	PCE (%)	Reference
FTO/SnO_2_/3C-SiC/KSnI_3_/NiO/C	1.399	17.72	89.99	22.31	This work
FTO/SnO_2_/3C-SiC/KSnI_3_/NiO/C	1.392	18.27	89.57	22.78	Ref. [27]

**Table 5 nanomaterials-15-00580-t005:** Optimized parameters of FTO, ETLs, HTL, and KSnI_3_ for the four optimized PSC structures.

Parameters	FTO	Al–ZnO	LiTiO_2_	ZnO	SnO_2_	KSnI_3_	rGO
Thickness (nm)	100	489	50	489	489	50	2670
N_A_ (cm^−3^)						8.33 × 10^20^	5.45 × 10^21^
N_D_ (cm^−3^)		10^15^	10^12^	10^16^	10^16^		

**Table 6 nanomaterials-15-00580-t006:** The V_oc_, J_sc_, FF, and PCE of the optimized PSC configurations.

PSC Structure	V_oc_ (V)	J_sc_ (mA/cm^2^)	FF (%)	PCE (%)
FTO/Al–ZnO/KSnI_3_/rGO/Se	1.44	21.28	90.15	27.60
FTO/LiTiO_2_/KSnI_3_/rGO/Se	1.30	21.25	90.01	24.94
FTO/ZnO/KSnI_3_/rGO/Se	1.43	21.26	90.70	27.62
FTO/SnO_2_/KSnI_3_/rGO/Se	1.53	21.25	91.27	29.67

**Table 7 nanomaterials-15-00580-t007:** The V_oc_, J_sc_, FF, and PCE of the optimized FTO/SnO_2_/KSnI_3_/rGO/Se when ideal R_s_, R_sh_, and temperature are assumed.

PSC Structure	V_oc_ (V)	J_sc_ (mA/cm^2^)	FF (%)	PCE (%)
FTO/SnO_2_/KSnI_3_/rGO/Se	1.54	22.04	89.76	30.44

**Table 8 nanomaterials-15-00580-t008:** Performance metrics of the optimized PSC structures and literature.

PSC Structure	PCE (%)	Reference
FTO/Al–ZnO/KSnI_3_/rGO/Se	27.60	This paper
FTO/LiTiO_2_/KSnI_3_/rGO/Se	24.94	This paper
FTO/ZnO/KSnI_3_/rGO/Se	27.62	This paper
FTO/SnO_2_/KSnI_3_/rGO/Se	30.44	This paper
FTO/TiO_2_/KSnI_3_/Spiro-OMeTAD/W	9.776	[22]
FTO/C_60_/KSnI_3_/PTAA/C	10.83	[23]
FTO/TiO_2_/KSnBr_3_/Cu_2_O/Au	8.05	[52]
FTO/F_16_CuPc/KSnI_3_/CuPc/C	11.91	[24]
FTO/ZnOS/KSnI_3_/NiO/C	9.28	[25]
FTO/ZnO/KSnI_3_/CuI/Au	20.99	[26]
FTO/SnO_2_/3C–SiC/KSnI_3_/NiO/C	22.78	[27]

**Table 9 nanomaterials-15-00580-t009:** The V_oc_, J_sc_, FF, and PCE of the optimized FTO/SnO_2_/KSnI_3_/rGO/Se when R_s_ = 0.5 Ω cm^2^.

PSC Structure	V_oc_ (V)	J_sc_ (mA/cm^2^)	FF (%)	PCE (%)
FTO/SnO_2_/KSnI_3_/rGO/Se	1.55	22.04	89.29	30.21

## Data Availability

All data presented in this article are available on request.

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
