# Peer review of "Modeling and Analysis of KSnI3 Perovskite Solar Cells Yielding Power Conversion Efficiency of 30.21%"

_nanomaterials, 2025, doi:10.3390/nano15080580_

Round 1

Reviewer 1 Report

Comments and Suggestions for Authors

This manuscript studied the KSnI3 by SCAPS-1D with rGO and WSe2 as transport layers, and the detailed research results were provided. In my opinion, the manuscript can be considered to publish in the Nanomaterials after some issues are figured out.

  1. The characteristics of KSnI3, such as electronics, optics, stability, etc., need to be summarized and explained in the article;
  2. The band value of KSnI3 needs to be explained, such as calculated by DFT using GGA functional, because this functional will underestimate the band value. Also, are there any experimental band values that can be provided in the manuscript?
  3. Why the electron affinity of KSnI3 is 3.44 eV? This requires a more detailed explanation as the electron affinity directly affects the simulated characteristics.

Author Response

Comment 1: The characteristics of KSnI3, such as electronics, optics, stability, etc., need to be summarized and explained in the article;

Response 1: We have now added the details on the properties of KSnI3. Please see the text that is highlighted in yellow in the introduction.

Comment 2: The band value of KSnI3 needs to be explained, such as calculated by DFT using GGA functional, because this functional will underestimate the band value. Also, are there any experimental band values that can be provided in the manuscript?

Response 2: The details on how the band gap value was obtained have been added as a footnote in table 1 and below table 2. Unfortunately, there is currently no experimental value that can be used for comparison. Thus, we used the value that was used in the recent study of Ref. [28] which showed the highest PCE in the literature so far, so that we could make direct comparison and investigate the effect of using different HTL and ETL materials, for example.

Comment 3: Why the electron affinity of KSnI3 is 3.44 eV? This requires a more detailed explanation as the electron affinity directly affects the simulated characteristics.

Response 3: The electron affinity of KSnI3 was obtained from the theoretical study of Ref. [23]. The details on how it was computed has now been added as footnote. Please the text highlighted in yellow in table 1 and below table 2. The approach was also used in other studies, please see Aditya Kumar et. Al, ACS Omega 2021, 6, 7086−7093 on https://dx.doi.org/10.1021/acsomega.1c00062

Reviewer 2 Report

Comments and Suggestions for Authors

The manuscript presents an interesting study on the simulation-based optimization of KSnI₃-based perovskite solar cells. The use of lead-free perovskites is an emerging area in photovoltaics that aligns with the growing demand for environmentally friendly photovoltaic technologies. The reported improvements in device efficiency through structural modifications are noteworthy. However, several aspects require further clarification and discussion before proceeding with the submission. Below are my comments and questions regarding the study.

  1. First, the quality of the graphs should be improved. It is recommended to modify graphs with lines and dots/points to improve readability and convenience for readers.

  1. The absorption coefficient equation (Eq. 6) uses constants (α = 10⁵, β = 10⁻¹²) generally used in SCAPS-1D simulations. How were these values determined, and do they accurately represent KSnI₃’s optical properties?

  1. The PCE of 30.44% is achieved under ideal conditions (Rₛ = 0, Rₕ = 10⁷ Ω·cm², T = 371 K). Why weren’t more realistic resistance values (e.g., Rₛ > 0) simulated, and how does this impact the practical significance of the results?

  1. Why does PCE increase with temperature up to 371 K (Fig. 15), contrary to typical PSC behavior? What physical mechanisms explain this, and how does KSnI₃’s stability hold at such high temperatures? Moreover, Tin-based perovskites like KSnI₃ are prone to oxidation. How do the proposed structures (e.g., rGO/Se interface) address this stability issue, especially at 371 K?

  1. There seems to be a minor contradiction in the sentence (line 209) regarding the effect of PL thickness. The first part states that if the PL is "too thick," it will not have sufficient volume to absorb light, which seems incorrect. Instead, should it be "too thin"? Please clarify and revise for consistency. Moreover, some spellings need to be corrected, such as in line 222 (add “field” after electric).

Author Response

Comment 1: First, the quality of the graphs should be improved. It is recommended to modify graphs with lines and dots/points to improve readability and convenience for readers.

Response 1: We have now improved all figures for readability by using lines with points, as well as grids in the x and y directions. Please see figure 4 to 16.

Comment 2: The absorption coefficient equation (Eq. 6) uses constants (α = 10⁵, β = 10⁻¹²) generally used in SCAPS-1D simulations. How were these values determined, and do they accurately represent KSnI₃’s optical properties?

Response 2: We have now corrected this part and added details on how α and β are calculated in the new version of SCAPS-1D, which was used in the study. Please see the text below equation 6.

Comment 3: The PCE of 30.44% is achieved under ideal conditions (Rₛ = 0, Rₕ = 10⁷ Ω·cm², T = 371 K). Why weren’t more realistic resistance values (e.g., Rₛ > 0) simulated, and how does this impact the practical significance of the results?

Response 3: We have added the case where we assume non-zero series resistance of 0.5 Ω·cm², which was suggested in the literature by Ref. [29]. Please see the text highlighted in green in section 3.13 and table 9. We also updated this in the conclusion, abstract, and title.

Comment 4: Why does PCE increase with temperature up to 371 K (Fig. 15), contrary to typical PSC behavior? What physical mechanisms explain this, and how does KSnI₃’s stability hold at such high temperatures? Moreover, Tin-based perovskites like KSnI₃ are prone to oxidation. How do the proposed structures (e.g., rGO/Se interface) address this stability issue, especially at 371 K?

Response 4: These details on the dependence of PCE on temperature have been added. Please see the green text in section 3.11. New Ref. [57] was also added. We have also added recommendation on how to suppress oxidation and improve efficiency and stability. Please see text highlighted in green in the last paragraph of section 3.13 and the corresponding references.

Comment 5: There seems to be a minor contradiction in the sentence (line 209) regarding the effect of PL thickness. The first part states that if the PL is "too thick," it will not have sufficient volume to absorb light, which seems incorrect. Instead, should it be "too thin"? Please clarify and revise for consistency. Moreover, some spellings need to be corrected, such as in line 222 (add “field” after electric).

Response 5: The typos have now been corrected. Please see the green text in the 2nd and 3rd paragraphs of section 3.

Reviewer 3 Report

Comments and Suggestions for Authors

The manuscript reports results of a computational study of various KSnI3 perovskite solar cell structures with the goal to optimize the conversion efficiency by fine tuning the thicknesses and dopant densities of the rGO, KSnI3, Al-ZnO, LiTiO2, ZnO, and SnO2. The paper claims that the highest efficiency goes beyond 30% and that the FTO/SnO2/KSnI3/rGO/Se device is 7.66% more efficient than the best performing device reported in the literature so far.  For the device simulations, the manuscript uses SCAPS-1D, a well known and widely used one-dimensional solar cell simulation tool developed at the University of Gent, Belgium.  

Although the report brings only incremental progress to the field and the literature is already flooded with SCAPS-1D simulations, the paper is still interesting.  I particularly like the comparison with other studies, which puts the present results in perspective.

Since the paper has a rather standard format, my main concerns regard the research design and the use of the software without clearly specifying its limits.

  1. In particular, although it attempts to push the limits of the conversion efficiency, the paper does not make any mention of the fundamental Shockley-Queisser approach, which puts a theoretical limit to the efficiency based on various loss mechanisms. Since any increase in efficiency is due to a decrease in losses, the paper should discuss in more detail what such lowered losses likely lead to the claimed 7% increase in efficiency.
  2. Secondly, the manuscript does not mention the limits of the software used. I particularly recommend the work of Saidarsan et al. (https://doi.org/10.1016/j.solmat.2024.113230), which mentions such limits and helps in promoting good research practice in the field.

2.1. In particular, the manuscript ought to address more clearly issues such as the optical losses, as SCAPS-1D focuses more on the charge transport and less on reflection losses at the several intermediate interfaces. Ignoring optical losses leads to overestimations of the overall efficiency.

2.2.  A second issue regards the optical absorption model for exciton generation, which is based on some assumptions that may not be applicable to the materials at hand.  Actual absorption profiles may possess intricate features, including peaks and troughs at specific wavelengths, which are intrinsic and characteristic of the material, that are not taken into account by the simple Eg-sqrt model used in SCAPS-1D.

2.3.  The discussion of the choice of the radiative recombination coefficient (Cr) is insufficient in the manuscript, which brings worries of using too low values of the coefficient. This may lead, again, to underestimations of losses and overestimations of efficiencies.

  1. The manuscript reports only changes of two types of parameters, the layer thicknesses and the dopant densities. While this choice is natural, these two types of parameters being very important, it is also rather limiting. I strongly recommend that a defect density optimization is also performed. It is a well-established fact that the defect concentration (vacancies, Frenkel defects, interstitials, Schottky defects, etc.) of the absorber, charge transport layers, and interfaces have a substantial impact on the SC performance and stability. Therefore, adding this type of optimization parameter may be crucial to the understanding of the loss mechanisms and to proposing the best possible device improvements.
  2. Finally, I have a couple of suggestions regarding form, not content. One refers to the caption of Fig. 3, which displays the gap between valence and conduction bands. The present caption suggests that the gap is filled with states, which is misleading. The other refers to the number of graphs. Many graphs are repetitive, lengthening the paper too much and preventing an easy comparison.  I suggest that the plots of the short-circuit current density, open-circuit voltage, fill factor and efficiency assemble the data from all devices, to facilitate the comparison.

Author Response

Comment 1: In particular, although it attempts to push the limits of the conversion efficiency, the paper does not make any mention of the fundamental Shockley-Queisser approach, which puts a theoretical limit to the efficiency based on various loss mechanisms. Since any increase in efficiency is due to a decrease in losses, the paper should discuss in more detail what such lowered losses likely lead to the claimed 7% increase in efficiency.

Response 1: We have now included the details on the Shockley-Queisser limit. Please see the text highlighted in orange in the 3rd paragraph of the introduction.

Comment 2: Secondly, the manuscript does not mention the limits of the software used. I particularly recommend the work of Saidarsan et al. (https://doi.org/10.1016/j.solmat.2024.113230), which mentions such limits and helps in promoting good research practice in the field.

Response 2: The limits of the SCAPS 1D have now been aded. Please see details in response 2.1 to 2.3.

Comment 2.1: In particular, the manuscript ought to address more clearly issues such as the optical losses, as SCAPS-1D focuses more on the charge transport and less on reflection losses at the several intermediate interfaces. Ignoring optical losses leads to overestimations of the overall efficiency.

Response 2.1: This part has been added. Please the text highlighted in orange below table 8. New Ref. [58] was also added.

Comment 2.2: A second issue regards the optical absorption model for exciton generation, which is based on some assumptions that may not be applicable to the materials at hand. Actual absorption profiles may possess intricate features, including peaks and troughs at specific wavelengths, which are intrinsic and characteristic of the material, that are not taken into account by the simple Eg-sqrt model used in SCAPS-1D.

Response 2.2: This part has also been added. Please the text highlighted in orange below table 8.

Comment 2.3: The discussion of the choice of the radiative recombination coefficient (Cr) is insufficient in the manuscript, which brings worries of using too low values of the coefficient. This may lead, again, to underestimations of losses and overestimations of efficiencies.

Response 2.3: Indeed users have an option to correct for radiative recombination in SCAPS-1D, but adjusting Cr which accounts for recombination of conduction electrons with valence band holes. However, as pointed out in the SCAPS-1D application note (https://scaps.elis.ugent.be/SCAPS%20Application%20Note%20Shockley-Queisser%20limit.pdf), for most practical materials the Shockely-Read Hall (SRH) recombination dominates the total recombination rate (in SCAPS simulation) and Cr becomes irrelevant, if the defect density Nt in the active layer is sufficiently high (Nt > 1011) which is the case in our study where we have Nt = 1015. For instance, with this reasoning, we are also able to reproduce experimental PCE of ITO/TiO2/CsSnBr3/Spiro-OMeTAD/Au which was also reproduced in https://doi.org/10.1021/acs.energyfuels.4c00953 . Thus, we applied the same thinking for our study.

Comment 3: The manuscript reports only changes of two types of parameters, the layer thicknesses and the dopant densities. While this choice is natural, these two types of parameters being very important, it is also rather limiting. I strongly recommend that a defect density optimization is also performed. It is a well-established fact that the defect concentration (vacancies, Frenkel defects, interstitials, Schottky defects, etc.) of the absorber, charge transport layers, and interfaces have a substantial impact on the SC performance and stability. Therefore, adding this type of optimization parameter may be crucial to the understanding of the loss mechanisms and to proposing the best possible device improvements.

Response 3: This part has now been added. Please the text highlighted in orange in section 3.12 and figure 16.

Comment 4: Finally, I have a couple of suggestions regarding form, not content. One refers to the caption of Fig. 3, which displays the gap between valence and conduction bands. The present caption suggests that the gap is filled with states, which is misleading. The other refers to the number of graphs. Many graphs are repetitive, lengthening the paper too much and preventing an easy comparison. I suggest that the plots of the short-circuit current density, open-circuit voltage, fill factor and efficiency assemble the data from all devices, to facilitate the comparison.

Response 4: The caption of figure 3 has been modified accordingly. Due to time constraints, we are unable to combine performance metrics figures to minimize the number of figures. However, We highly appreciate this suggestion and if the reviewer insist, we are willing to ask for more time from editors so that we can combine the figures.

Round 2

Reviewer 3 Report

Comments and Suggestions for Authors

The revised manuscript has addressed the main concerns formulated in the first report and is significantly improved compared to the initial manuscript. In its present form the paper is almost ready for publication.  The authors mentioned in their response that they don’t oppose reducing the number of repetitive plots but that time constraints have preventing them to actually perform it. Now it remains to the editor to consider whether the suggestion to assemble the plots of the short-circuit current density, open-circuit voltage, fill factor and efficiency, to facilitate the comparison is important or not.  

Author Response

Comment: The revised manuscript has addressed the main concerns formulated in the first report and is significantly improved compared to the initial manuscript. In its present form the paper is almost ready for publication. The authors mentioned in their response that they don’t oppose reducing the number of repetitive plots but that time constraints have preventing them to actually perform it. Now it remains to the editor to consider whether the suggestion to assemble the plots of the short-circuit current density, open-circuit voltage, fill factor and efficiency, to facilitate the comparison is important or not.

Response: We have now combined Jsc and Voc in one panel, as well as PCE and FF in one panel, for each section. The length of the paper has greatly decreased by 5 pages. Please see attached version version.  
